# Somatosensory Intervention Targeting Temporomandibular Disorders and Awake Bruxism Positively Impacts Subjective Tinnitus

**DOI:** 10.3390/audiolres15050114

**Published:** 2025-09-09

**Authors:** Eric Bousema, Pieter U. Dijkstra, Pim van Dijk

**Affiliations:** 1Head & Neck Surgery, Department of Otorhinolaryngology, University Medical Centre Groningen, University of Groningen, 9700 RB Groningen, The Netherlands; p.van.dijk@umcg.nl; 2Het Kaak Hoofd Centrum, 6131 AL Sittard, The Netherlands; 3Department of Rehabilitation Medicine, University Medical Centre Groningen, University of Groningen, 9713 GZ Groningen, The Netherlands; 4Sirindhorn School of Prosthetics and Orthotics, Faculty of Medicine, Siriraj Hospital, Mahidol University, Bangkok 10700, Thailand; 5Graduate School of Medical Sciences (Research School of Behavioural and Cognitive Neurosciences), University of Groningen, 9713 AV Groningen, The Netherlands

**Keywords:** tinnitus, physical therapy modalities, temporomandibular joint disorders, pain, bruxism, treatment outcome

## Abstract

**Objective:** To analyze the effects of a somatosensory education intervention targeting temporomandibular disorders (TMD) and awake bruxism on subjective tinnitus. **Methods:** This study had a pre-post-design in a primary care practice for orofacial physical therapy. Twenty-eight participants with the presence of TMD and suffering from moderate to severe subjective tinnitus, for at least 3 months, received the following treatments: (a) comprehensive information about tinnitus and the factors influencing it; (b) bruxism reversal training via a smartphone application; and (c) treatment for TMD. The primary outcome was the Tinnitus Functional Index (TFI). Secondary outcomes were awake bruxism frequency and the TMD pain screener. The study was approved by the Ethics Committee of the University of Groningen, the Netherlands. **Results:** The mean (95% CI) reduction in TFI scores and awake bruxism frequency were 18.4 (13.2–23.5) and 16.6% (2.0–31.2%), respectively. A clinically relevant reduction of 13 points on the TFI was observed in 63% of the participants. Regression analysis revealed that factors associated with TFI change included the TFI initial score at T0 (0.3, 95% CI 0.0–0.6), the presence of daytime clenching (21.0, 95% CI 8.7–33.4), and stiffness or pain around the TMJ (10.6, 95% CI −1.9–23.0) at baseline. **Conclusions:** The findings suggest that tinnitus education, TMD treatment, combined with decreasing awake bruxism, can reduce tinnitus in a primary care setting.

## 1. Introduction

Subjective tinnitus is the perception of sound without the presence of an external source [1]. It is a heterogeneous and complex condition affecting 5–42% of the general population, of which 3–30% report their tinnitus as severe [2]. Tinnitus adversely impacts the quality of life in 1–4% of the general population, rendering it a considerable medical and socioeconomic challenge [3,4].

At present, there is no general consensus on the pathophysiological explanation for tinnitus. However, the most common explanation is that tinnitus is associated with aberrant spontaneous neural activity [1,5]. Many current treatments, therefore, cannot directly treat the cause of tinnitus and instead focus on factors that can modulate its severity.

Current conservative interventions for tinnitus relief can be categorized as: sound enrichment (hearing aids, sound therapy) [6], cognitive behavioral therapy [7], desensitization of the central nerve system by means of relaxation therapy and meditation [8,9], and somatosensory interventions such as treatment of temporomandibular disorders (TMD) [10]. Patient personalized combinations of these interventions may be more effective than a single intervention [11].

Patients with tinnitus have an increased risk of TMD, due to a bidirectional association between TMD and tinnitus [12,13,14]. About two-thirds of the patients with subjective tinnitus can modulate the loudness and pitch of their tinnitus by contracting neck, head, or jaw muscles [15,16,17,18,19,20,21,22]. Due to a bidirectional association, it can be hypothesized that treatment of jaw-related complaints may also provide relief of tinnitus.

The typical approach to treating TMD involves education, massage of jaw muscles, stretching of neck muscles, and splint therapy for reducing sleep bruxism. TMD is associated with bruxism [23,24,25]. In addition to sleep bruxism, there is also awake bruxism, in which the patient semi-voluntarily clenches the teeth or braces the mandible without tooth contact when awake. Symptoms of TMD are more commonly observed in individuals with awake bruxism than with sleep bruxism [26]. Therefore, reducing awake bruxism, in particular, should receive attention in the treatment of TMD and tinnitus [27].

In randomized controlled trials, in primary and tertiary care, small beneficial effects were found for TMD treatment in patients with tinnitus [28,29,30]. Studies investigating the effects of treatment of TMD and awake bruxism for patients with tinnitus in primary care have not been conducted. If such a treatment in primary care is effective in reducing tinnitus, it would increase accessibility to treatment.

The aim of this study was to investigate the effects of a somatosensory intervention targeting TMD and awake bruxism on tinnitus.

## 2. Methods

### 2.1. Study Design

This study had a pre-post-design in a primary care practice for orofacial physical therapy. The study was approved by the Ethics Committee of the University of Groningen, the Netherlands (UMCG RR number: 2019004550). The study was conducted between November 2019 and May 2021.

### 2.2. Participants

Potential participants with subjective tinnitus and TMD were recruited through flyers and word-of-mouth. Inclusion criteria were the presence of TMD and suffering from moderate to severe subjective tinnitus, for at least 3 months, as documented by a score on the Tinnitus Functional Index (TFI) between 25 and 90 points [31]. Additionally, participants must have access to a smartphone. Participants were excluded if tinnitus was associated with an otological or neurological disease such as Menière’s disease, progressive middle ear pathology, intracranial pathology such as hematoma, a tumor, or an abscess. Participants with previous surgery or trauma, such as fractures in the orofacial area, or those who underwent another treatment for their tinnitus during the study, were also excluded. Participants who met the inclusion criteria were invited to the physical therapy practice for TMD treatment and to participate in the study. Prior to the study, potential participants were informed, verbally and in writing, by the primary researcher (EB) concerning the study and its procedures. Written informed consent was obtained from all participants before entering the study. Participants were asked to complete a number of questionnaires, including questions about tinnitus and TMD characteristics, the TFI, and the TMD pain screener [32]. After completion of the treatment, assessments were repeated. Two additional open-ended questions were asked to gauge the participants’ perception of whether the treatment had improved their tinnitus and bruxism.

### 2.3. Treatment

The treatment consisted of three elements. First, participants received comprehensive information about tinnitus and the factors influencing it (such as stress, negative cognitions, and TMD). They were also informed that the goal of the treatment was to reduce tinnitus, making it easier for them to cope with.

The second element involved bruxism reversal training using a smartphone application, BruxApp^®^ (Version 2.5.8, BruxApp team, Pontedera, Italy). Participants were educated on how to use the BruxApp. The training aimed to reduce awake bruxism. As a result of this training, participants began identifying specific circumstances and situations in which they engaged in bruxism. These circumstances were then discussed during treatment sessions for educational purposes.

Third, during 30 min sessions once a week, participants received evidence-based TMD therapy [33] including therapist-assisted mobilization of the temporomandibular joint, manual treatment of trigger points, and instructions on self-care, such as jaw exercises, stretching, and relaxation techniques. In consultation with the dentist, splint therapy was applied. Treatment stopped when no further reduction in symptoms occurred.

### 2.4. Assessment Instruments

#### Primary Outcome Measure

The primary outcome was the score on the Tinnitus Functional Index (TFI). The TFI assesses tinnitus severity. It consists of 25 questions divided into eight subscales. Each question is answered on a scale from 0 to 10. The final outcome is normalized to a range from 0 to 100, with a higher score indicating a greater level of tinnitus severity. The TFI demonstrates good test–retest reliability (r = 0.78). Additionally, it has a good convergent validity with the Tinnitus Handicap Inventory (THI) (r = 0.86) and the Visual Analogue Scale (VAS) (r = 0.75). A change of 13 points is considered clinically relevant [31,34].

### 2.5. Secondary Outcome Measures

**Awake bruxism frequency.** Awake bruxism was measured as the number of times per day that the participants engaged in bruxism when awake. The BruxApp smartphone application was used for signaling and treatment of awake bruxism.

The app registered five conditions: relaxed jaw muscles, teeth contact, teeth clenching, jaw clenching, and teeth grinding. The application was programmed to send 20 sound alerts during the day. The alerts were sent at random intervals to reduce expectation bias. The participant could choose the type of sound alert to distinguish it from other notifications.

To assess awake bruxism frequency, the participant had to report their oral condition at each sound alert for six consecutive days [35,36]. Recording time was set from 8:00 to 12:30 and from 14:30 to 22:00 to allow a break during lunchtime [37]. After 6 days, participants continued receiving only sound alerts to monitor oral conditions, which helped them to unlearn the awake bruxism. Each time a person recognized bruxism behavior during an alert, they were instructed to stop their bruxism and conduct a simple and short relaxation or mobilization exercise for the neck and jaw. After a while, participants reported not needing the BruxApp anymore and started performing exercises on their own when they recognized bruxism behavior without the alerts.

**TMD pain screener**. The TMD pain screener consists of six questions and is used to assess TMD pain. If a participant scores positively on at least three out of the six questions, it suggests that the participant is experiencing painful TMD. The sensitivity of the screener is 0.99, and the specificity ranges from 0.95 to 0.98 [32].

### 2.6. Sample Size Calculation

We used a continuous response variable. Based on prior data, we assumed that the difference in the TFI response would be normally distributed with a standard deviation of 24.1. The difference in the mean response was expected to be 13 [31]. To detect this difference with 80% power and reject at a significance level of 0.05 (α = 0.05), a total of 29 subjects would be needed.

### 2.7. Participant Satisfaction

At follow-up, participants were asked an open-ended question about what the treatment had done for them. A summary of their answers will be presented.

### 2.8. Statistical Analysis

Statistical analyses were performed using SPSS software version 28.0.1.0 (IBM, Armonk, NY, USA). To test whether dropouts differed from the remaining participants, baseline characteristics were compared using the Mann–Whitney U or Chi-squared test. The remaining data were checked for normal distribution using the Kolmogorov–Smirnov and Shapiro–Wilk tests. If data met the statistical assumption of homogeneity of variance, a paired *t*-test was applied; otherwise, the Wilcoxon-signed rank test was applied. A *p*-value ≤ 0.05 was considered statistically significant. A regression analysis was performed with TFI change (ΔTFI) as the dependent variable. ΔTFI was calculated by subtracting the TFI score at baseline from that at follow-up. So a negative value indicates improvement in tinnitus. Factors associated with ΔTFI (*p* < 0.15) were included for the multivariable regression analyses.

## 3. Results

### 3.1. Subjects

A total of 43 subjects with self-reported tinnitus symptoms were screened for eligibility between November 2019 and June 2021. Thirty-nine participants met the inclusion criteria and agreed to participate. During the treatment period, ten individuals dropped out. Four participants began alternative tinnitus treatments, two participants had influenza, one sustained a fall, one participant no longer received reimbursement for the treatment, and another was advised by a gnathologist to discontinue the treatment. One participant dropped out because of a temporary increase in symptoms. Data from one participant was excluded from analysis, as this person filled out the baseline TFI questionnaire exclusively with scores of 0, which was interpreted to be unreliable. Overall, 28 participants completed treatment and both pre- and follow-up questionnaires (Table 1).

No significant differences in participant characteristics were found between dropouts and the study group on age (*p* = 0.79) or gender (*p* = 0.27). TFI (*p* = 0.68), awake bruxism frequency (*p* = 0.56), clenching the jaw during the day (*p* = 0.25), and stiffness or pain around the TMJ (*p* = 0.58). During treatment, no changes in medication use were reported by the patients. The median number of weekly treatment sessions was 9.0 (Table 1).

### 3.2. TFI Responses to Treatment

The mean TFI scores at baseline and at the completion of treatment met the statistical assumption of homogeneity of variance, allowing the use of a paired *t*-test. The results showed a significant reduction in the TFI score. A clinically relevant reduction of at least 13 points occurred in 63% of the participants (Figure 1, Table 2 and Appendix A). 

### 3.3. Awake Bruxism Responses to Treatment

The paired *t*-test revealed a statistically significant reduction of 16.6% (Table 2). Due to updates from Apple or Android, there were periods when the BruxApp malfunctioned. As a result, follow-up measurements could not be conducted for 13 individuals, leaving data from 15 individuals for analysis.

### 3.4. Pain Responses to Treatment

The scores on the TMD pain screener did not change significantly between baseline and post-treatment (Table 2).

### 3.5. Factors Associated with Change in TFI Reduction

In the univariate regression analysis, factors associated with ΔTFI (*p* < 0.15) included gender, age, TFI (T0), reduction in awake bruxism, changes in tinnitus pitch, modularity of tinnitus, pain or stiffness around the TMJ, and clenching during the daytime. Factors that seemed to have no clinical effect on the degree of tinnitus reduction, such as the number of treatment sessions, were not included in the univariate analysis. The ΔTFI-associated factors were included in the multivariable regression analysis. In this analysis, the TFI score at T0, presence of daytime clenching, and presence of stiffness or pain at T0 in the TMJ area were significantly associated with ΔTFI (explained variance = 67%) (Table 3).

### 3.6. Participant Satisfaction

A summary of participant responses to an open evaluation question showed a wide range of positive outcomes following treatment. Many have experienced a reduction in tinnitus, especially during jaw relaxation exercises and awareness of stress-related factors. Although not all results were equally strong, the treatment generally provided the participants with a better understanding of tinnitus and offered techniques to reduce symptoms.

## 4. Discussion

In this study, we investigated the impact of a somatosensory intervention combined with education targeting TMD and reducing awake bruxism on tinnitus within a primary care setting. The results showed TFI scores had reduced by an average of 18.4 points. This change exceeds 13 points, which is considered to be a clinically relevant change [31]. Awake bruxism also reduced significantly (mean reduction: 16.6%). Participants who clenched their jaw or reported pain/stiffness in the TMJ area at baseline showed a greater reduction in TFI at follow-up.

Our results are in line with other studies that have analyzed the effects of TMD treatment on tinnitus. In one study, a reduction of 13.1 points on the THI questionnaire score was found after six sessions of TMD treatment [28]. In a multidisciplinary study (with otorhinolaryngologists, audiologists, physical therapists, and dentists) [29], TFI scores decreased on average by 13.8 points after nine weeks of TMD treatment. The waiting list group showed a decrease of 5.0 points. In an observational study, 83% and in another study 62% of the tinnitus symptoms resolved partially or completely after TMD treatment [30,38].

All of the above studies pay attention to reducing awake bruxism, but not as intensively as with continuous monitoring using the BruxApp. Adding continuous monitoring to our treatment might explain our slightly larger treatment effect [39]. However, based on our results, we cannot demonstrate that a reduction in awake bruxism or the use of the BruxApp contributed to the reduction of tinnitus. Although the frequency of awake bruxism decreased significantly, regression analysis showed that this reduction was not associated with a decrease in the TFI. The latter result may be related to the poor functionality of the BruxApp.

The missing 13 follow-up BruxApp measurements due to continuous updates from Apple and Android are a point of concern. Often, the app functioned well, but there were periods after updates when it stopped recording after a couple of days. Patients had to restart, but after several attempts, they gave up. Although awake bruxism decreased in 79% of the patients, as assessed through an evaluation question, the presence of missing data introduces a risk of bias. Therefore, we should be cautious in drawing the conclusion that the reduction in TFI can be directly attributed to a decrease in clenching behavior.

Another matter of concern in this study was the dropout rate. A total of 25% of the participants did not complete the study. One participant discontinued due to a short-term increase in tinnitus symptoms. However, the reasons for dropout among the other participants were unrelated to the treatment. Analysis showed no significant differences in the characteristics of the dropout group compared to those who completed the study.

Our study is a pilot study without a control group, conducted in preparation for an RCT; hence, we cannot rule out a placebo effect of the treatment. In a meta-analysis, it was found that the placebo effect of tinnitus treatments was on average 5.6 (95% CI 3.3–8.0) points for the Tinnitus Handicap Inventory and 2.9 (95% CI −2.8–8.5) points for the TFI [40]. Although different treatments presumably have different placebo effect magnitudes, it seems reasonable to assume that our treatment effect, 18.4 (95% CI 13.2–23.5) points, exceeds the placebo effect. Additionally, in a large longitudinal study, 82% of the patients reported still experiencing bothersome tinnitus after 4 years. The median tinnitus duration in our study was 3.5 years, making it unlikely that the improvement can solely be attributed to natural recovery [41].

The intriguing aspect of this study is that, in general, focusing on tinnitus increases the severity of symptoms. In this study, such an increase did not occur during treatment. By focusing treatment on TMD, there was no emphasis on reducing tinnitus symptoms, which is a strength of our treatment. Nevertheless, there was a clear decrease in the tinnitus burden. Another strength is that this treatment was administered in a primary care setting, making it probably more cost-effective and easier for patients to access close to home, compared to secondary or tertiary care. On the other hand, otologic examination, such as tonal audiometry, cannot be performed in a physical therapy practice and requires collaboration with other disciplines to identify hearing loss.

TMD treatment, combined with the reduction in awake bruxism and education, can reduce tinnitus in a primary care setting. Continuous monitoring of bruxism and discussing specific bruxism situations with the patient could contribute to these improvements. A further RCT into TMD treatment for patients with tinnitus is necessary to enhance understanding and improve treatment effectiveness. This RCT should include a follow-up after the end of the treatment to determine the long-term effects and possibly different intervention groups to investigate which of the various components of this treatment is effective.

## 5. Conclusions

The findings suggest that tinnitus education, TMD treatment, combined with decreasing awake bruxism, can reduce tinnitus in a primary care setting.

## Figures and Tables

**Figure 1 audiolres-15-00114-f001:**
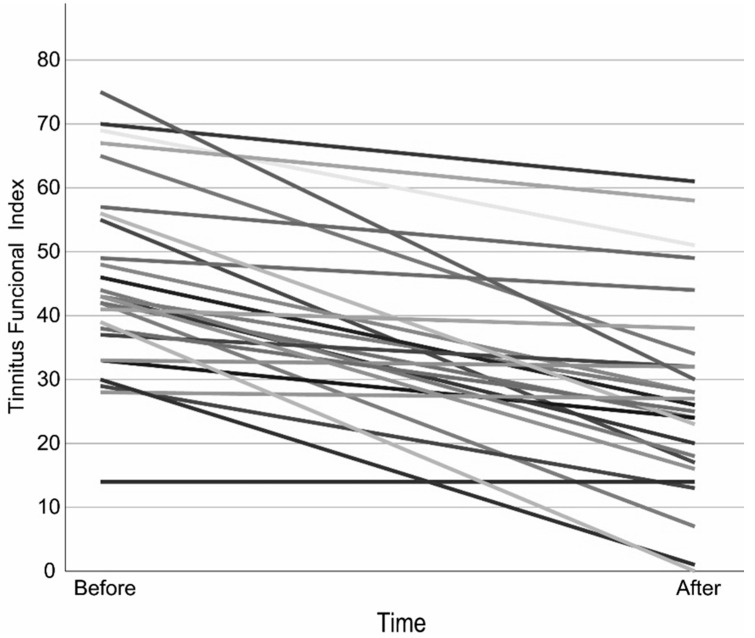
Tinnitus Functional Index over time per Participant. Each line represents one subject, to distinguish each subject different shades are applied.

**Table 1 audiolres-15-00114-t001:** Participant characteristics.

Baseline Variable	
Gender (male/female)	13/15
Median Age y (IQR)	55.8 (26.3–74.9)
Median tinnitus duration y (IQR)	3.5 (0.2–42.9)
Median number of treatment sessions (IQR)	9.0 (7.0–10.0)
% of participants who report:	
Tinnitus	
Pitch modulation during the day	35.7%
Loudness modulation during the day	71.4%
Modulation through jaw or neck movements	60.7%
Bruxism	
Nighttime teeth grinding	38.1% (8/21)
Clenching the jaw during the day	75.0% (18/24)
Pain	
Pain in the neck	89.2%
Stiffness or pain in the TMJ area	70.4%

**Table 2 audiolres-15-00114-t002:** Primary and secondary outcomes at baseline and follow-up.

	Baseline Mean (SD)	Follow Up Mean (SD)	Change (95% CI)	*p*
Tinnitus functional index	45.8 (14.6)	27.4 (15.5)	−18.4 (13.2 to 23.5)	<0.001 *
Awake bruxism frequency	58.0% (31.2%)	41.4% (30.2%)	−16.6% (2.0–31.2%)	0.01 *
TMD pain screener	1.8 (2.2)	1.5 (2.0)	−0.3 $	0.80 #

Awake bruxism frequency: mean percentage of awake bruxism manifestations during notifications. TMD: temporomandibular disorders. SD: standard deviation. 95% CI: 95% confidence interval. * results of paired *t*-test. # results of the Wilcoxon Signed Ranks Test. $: No confidence intervals provided.

**Table 3 audiolres-15-00114-t003:** Results of multivariable regression analyses with TFI change as the dependent variable.

	Regression Coefficient	SE	t	*p*-Value	95% CI
TFI T0	0.3	0.1	2.3	0.054	[0.0–0.6]
Clenching the jaw during the day	21.0	5.4	3.9	0.004	[8.7–33.4]
Stiffness or pain in the TMJ area	10.6	5.4	2.0	0.086	[−1.9–23.0]
Intercept	−18.7	8.5	−2.3	0.060	[−38.4–1.0]

SE = standard error.

## Data Availability

The data that support the findings of this study are available from the corresponding author (EB), upon reasonable request.

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
