# Peer review of "Somatosensory Intervention Targeting Temporomandibular Disorders and Awake Bruxism Positively Impacts Subjective Tinnitus"

_audiolres, 2025, doi:10.3390/audiolres15050114_

Round 1
Reviewer 1 Report
Comments and Suggestions for Authors
Tinnitus is a health problem with unclear ethiology and lack of effective treatment. Research aiming to solve these problems is therefore of high interest. The present study adress the connection between TMD and tinnitus and the possible effect on tinnitus through TMD treatment. Unfortunately the BruxApp had some functional problems and together with dropouts reducing the sample size and possibly affecting the results. The authors have addressed these problems and the authors need to be very modest regarding the method effectiveness against tinnitus. Nevertheless, significant reduction in TFI in combination with reduced bruxism was observed.
However, the duration and amount of treatment is not reported "Treatment stopped when no further reduction in symptoms occurred". What was the treatment span? Number of therapy sessions? Was there any connection between the amount of treatment and effect on symptoms? What was the effect of only information - control group missing.
Author Response
Dear Reviewer,
Thank you very much for taking the time to review this manuscript. Please find the detailed responses below and the corresponding revisions/corrections highlighted/in track changes in the attached file.
Your sincerely
Eric Bousema

Reviewer 2 Report
Comments and Suggestions for Authors
The manuscript is a pilot study conducted in preparation for an
RCT and it is focused on the efficacy of TMD and awake bruxism treatment in somatosensory tinnitus. Some points must be clarified and improved with a revision.
Line 85: did you mean otological?
Did you perform tonal audiometry?If yes, could you specify how many patients had normal hearing and/or hearing loss? If not I suggest to indicate it as a limit of the study and consider it for the next RCT planned.
89.2% of patients reported pain in the neck: the patients were taking medications for pain During the treatment patients were taking medications for other diseases/comorbidities?
Did you consider the possible co-occurence of obstructive sleep apnea syndrome (OSAS), insomnia, gastroesophageal reflux disease (GERD)? If not this is a limit of the study and can be considered for the next RCT planned.
Patients undergoing treatment with occlusal splint or other oral appliances, TENS, behavioural or pharmacological therapy were excluded or included?
Which was the average duration of treatment at the end of which the patients were re-evaluated?
Have you considered performing a follow-up after the end of treatment? If not this is a limit of the study and can be considered for the next RCT planned.
Author Response

(The authors gave the same response as above.)

Reviewer 3 Report
Comments and Suggestions for Authors
This is an interesting study exploring somatosensory tinnitus and temporomandibular disorders (TMD), particularly the phenomenon of awake bruxism. Study design is good with appropriate outcome measures and statistical tests. A suitable sample size calculation was performed. The manuscript is generally well written and easy to read. I only have a few relatively minor concerns.
The intervention was a complex intervention with three components: a tinnitus education component; bruxism reversal training using a smartphone app; TMD therapy. Throughout the manuscript, the education component has largely been ignored. In the title, there is no mention of the educational component. In the abstract the intervention is called “a somatosensory intervention targeting temporomandibular disorders and awake bruxism” (lines 16 – 17) and the positive benefits seen in the study are attributed to “TMD treatment, combined with the decreasing awake bruxism” (lines 31 – 32). In the discussion, the intervention is called “a somatosensory intervention targeting TMD and reducing awake bruxism” (lines 242 – 243) and success is attributed to “TMD treatment, combined with the reduction of awake bruxism, can reduce tinnitus in a primary care setting” (lines 292 – 293). In the conclusion it is stated that “TMD treatment, combined with decreasing awake bruxism can reduce tinnitus in a primary care setting” (lines 298 – 299). All of these statements ignore any improvement that might be attributed to the education component. From the study design it is impossible to infer which component has led to improvement and this needs to be reflected throughout the text. One could even play devil’s advocate here and suggest that as the tinnitus outcome measure improved but the TMD measure did not, it was more likely to be the tinnitus education module that led to improvement!
Lines 67 – 68. This sentence states “If such a treatment in primary care is effective in reducing tinnitus it would increase accessibility of treatment.” This would probably be better written as: If such a treatment in primary care is effective in reducing tinnitus, it could increase accessibility to treatment as it would reduce the need for secondary and tertiary healthcare services.
Line 85. Ontological should be otological (ontological refers to the philosophical nature of being and existence).
Lines 150 – 152. This sentence states “Based on prior data, we assumed that the difference in the response would be normally distributed with a standard deviation of 24.1.” This would be better written as: Based on prior data, we assumed that the difference in the TFI response would be normally distributed with a standard deviation of 24.1.
Line 178. “The flu” is a colloquialism and would be better written in a scientific article as influenza.
Figure 1 gives a good pictorial representation of the treatment outcomes but it is difficult to follow individual patients, especially in the middle of the chart. I would like to see a table of these data, either presented within the manuscript or as a supplementary file.
Lines 283 – 284. This sentence states “Additionally tinnitus duration was almost 3.5 years and natural recovery of tinnitus is unlikely after such a period.” The figure 3.5 years was the median tinnitus duration, and the patient with the shortest tinnitus duration had their symptoms for only 0.2 years which is certainly within the range where spontaneous resolution might occur. Furthermore, if you are going to claim that tinnitus does not improve after a certain time period, this should be supported with one or more appropriate references.
Lines 289 – 291. This sentence states “Another strength is that this treatment was administered in a primary care setting, making it probably more cost-effective and more easily available for participants in their own environment, compared to tertiary care.” This would probably be better written as: Another strength is that this treatment was administered in a primary care setting, making it probably more cost-effective and easier for patients to access close to home, compared to secondary or tertiary care.
There has not been much research regarding awake bruxism and tinnitus but a Brazilian group has published two papers on the topic:
Spisila T, Fontana LC, Hamerschmidt R, de Cássia Cassou Guimarães R, Hilgenberg-Sydney PB. Phenotyping of somatosensory tinnitus and its associations: An observational cross-sectional study. J Oral Rehabil. 2024 Oct;51(10):2008-2018. doi: 10.1111/joor.13783.
D'Amato ACOG, Spisila T, Camargo GS, Hilgenberg-Sydney PB. Frequency of Oral Behaviors as a Risk Factor for Somatosensory Tinnitus. Int Arch Otorhinolaryngol. 2025 Apr 15;29(2):1-5. doi: 10.1055/s-0045-1802577.
One or both of these papers should probably be referenced and discussed in this manuscript.
Author Response

(The authors gave the same response as above.)
